# Unraveling the Mechanisms Involved in the Beneficial Effects of Magnesium Treatment on Skin Wound Healing

**DOI:** 10.3390/ijms25094994

**Published:** 2024-05-03

**Authors:** Yuta Yoshino, Tatsuki Teruya, Chika Miyamoto, Mai Hirose, Satoshi Endo, Akira Ikari

**Affiliations:** Laboratory of Biochemistry, Department of Biopharmaceutical Sciences, Gifu Pharmaceutical University, Gifu 501-1196, Japan; yoshino-yu@gifu-pu.ac.jp (Y.Y.); 175076@gifu-pu.ac.jp (T.T.); 195119@gifu-pu.ac.jp (C.M.); 195105@gifu-pu.ac.jp (M.H.); sendo@gifu-pu.ac.jp (S.E.)

**Keywords:** keratinocyte, magnesium, matrix metalloproteinase, migration, wound healing

## Abstract

The skin wound healing process consists of hemostatic, inflammatory, proliferative, and maturation phases, with a complex cellular response by multiple cell types in the epidermis, dermis, and immune system. Magnesium is a mineral essential for life, and although magnesium treatment promotes cutaneous wound healing, the molecular mechanism and timing of action of the healing process are unknown. This study, using human epidermal-derived HaCaT cells and human normal epidermal keratinocyte cells, was performed to investigate the mechanism involved in the effect of magnesium on wound healing. The expression levels of epidermal differentiation-promoting factors were reduced by MgCl_2_, suggesting an inhibitory effect on epidermal differentiation in the remodeling stage of the late wound healing process. On the other hand, MgCl_2_ treatment increased the expression of matrix metalloproteinase-7 (MMP7), a cell migration-promoting factor, and enhanced cell migration via the MEK/ERK pathway activation. The enhancement of cell migration by MgCl_2_ was inhibited by MMP7 knockdown, suggesting that MgCl_2_ enhances cell migration which is mediated by increased MMP7 expression. Our results revealed that MgCl_2_ inhibits epidermal differentiation but promotes cell migration, suggesting that applying magnesium to the early wound healing process could be beneficial.

## 1. Introduction

The skin is the largest organ in the human body, which not only feels warmth and stimuli but also functions as a physical barrier between the internal and external sides of the organism, protecting against water loss from the body and the invasion of pathogens from the outside [1]. The skin has a three-layer structure consisting of the epidermis, dermis, and subcutaneous adipose tissue. The epidermis is the thickest layer containing keratinocytes and melanocytes, and it comprises four layers including the basal layer, spinous layer, granular layer, and stratum corneum [2]. Keratinocytes, which divide in the basal layer, lose their mitotic potential and differentiate as they move to the upper epidermal layers. After reaching the stratum corneum, the cells that have completed their final differentiation serve as a coating and protection for the body surface and eventually detach from the body surface [3].

When the skin is damaged physically, a complex cellular response called “wound healing” is initiated to maintain homeostasis by restoring the injured area to its pre-injury state. The process of skin wound healing is divided into in four stages as follows: the hemostatic, inflammatory, proliferative, and remodeling stages [4]. In the hemostatic stage, platelets assemble the surrounding blood cells to form a fibrin clot, covering the wounded region. Platelets then release growth factors and cytokines such as platelet-derived growth factors, leading to migration of the immune system cells and fibroblasts [4]. Subsequently, within hours of formation of the covered wound, the inflammatory phase involves exudate and infiltration by the immune system cells [5]. The immune system cells produce proteases and reactive oxygen species to defend the wound area against external contaminants including bacteria [6]. In addition, growth factors and cytokines over-produced by immune system cells enhance cell migration and the proliferation of keratinocytes and fibroblasts at the wound edge [7]. Epidermal keratinocytes migrate according to the wound edge by degrading the extracellular matrix through the effect of plasminogen [8] and matrix metalloproteinases (MMPs) [9]. Fibroblasts proliferate in the wound area, leading to the production of extracellular matrix components, such as collagen, fibronectin, elastin, and hyaluronan and the formation of granulation tissue containing nerve fibers and capillaries. In the remodeling phase, epidermal keratinocytes differentiate to form the stratum corneum [10]. Repeated synthesis and catabolism of collagen in the granulation tissue result in a maturing scar and spontaneous regression of the granulation tissue back to its pre-injured state [7].

Delayed inflammatory and proliferative phases increase the risk of developing bacterial skin infections, and delayed epidermal differentiation during the remodeling stage also reduces patient quality of life by causing pain and wound appearance problems such as keloids and hypertrophic scars [11,12]. Thus, each stage in the wound healing process is regulated strictly. Factors that delay wound healing include diabetes, nutritional deficiencies, immunodeficiency, aging, and bacterial infection in the wound area.

The magnesium ion (Mg^2+^) is an essential metal ion in the human body, and Mg^2+^ homeostasis is vital to maintaining the human health [13]. Mg^2+^ concentrations in vivo are regulated within a narrow range of 0.7–1.0 mM in healthy individuals. These concentrations are maintained by uptake from the gastrointestinal tract and excretion from the kidneys [13,14]. Mg^2+^ has a diverse range of physiological activities, alone or in concert with biomolecules. For example, Mg^2+^ is a cofactor for approximately 600 enzymes and an activator for more than 200 enzymes [13]. The hydrolysis of ATP by ATPases requires the complex formation of Mg^2+^ and ATP [15]. Therefore, Mg^2+^ is essential for various cellular responses including cell proliferation, cell cycle, and protein synthesis by both direct and indirect functions [16,17,18].

We reported that Mg^2+^ treatment of epidermal keratinocytes increases the expression of hyaluronic acid synthase via glycogen synthase kinase 3 and the cyclic AMP response element binding protein, resulting in increased hyaluronic acid production, which promotes wound healing [19]. We also demonstrated that Mg^2+^ treatment increases polyamine synthesis and reduces ultraviolet (UVB)-induced cell death in keratinocytes [20]. The above reports suggest that Mg^2+^ treatment may have several beneficial effects on epidermal keratinocytes. In addition, other researchers have reported that Mg^2+^ treatment promotes skin wound healing [21,22]. For example, treatment of hairless mouse skin wound sites with magnesium chloride (MgCl_2_) promotes skin barrier function repair [21]. Applying a hydrogel releasing Mg^2+^ and Zn^2+^ into the back skin of mice promotes wound healing by promoting extracellular matrix reorganization via a fibroblast epithelial–mesenchymal transition [22]. From the above studies, it can be seen that Mg^2+^ promotes cutaneous wound healing and thus has valuable applications in wound care technology. However, it needs to be fully understood which wound healing processes require Mg^2+^.

In this study, we investigated the effects of MgCl_2_ treatment on cell migration and epidermal differentiation using the human epidermal keratinocyte cell line HaCaT cells and primary cultured normal human epidermal keratinocyte (NHEK) cells.

## 2. Results

### 2.1. Effect of Mg^2+^ on the Expression Levels of Epidermal Differentiation-Promoting Factors

To clarify the effects of MgCl_2_ treatment on epidermal differentiation, differentiation-promoting factors, Tumor protein p63 (TP63), Desmoglein 3 (DSG3), Late Cornified Envelope 1E (LCE1E), FOS Like 2 (FOSL2), Wnt Family Member 5A (WNT5A), Semaphorin (SEMA) 3C, and SEMA5A mRNA expression levels were analyzed using real-time polymerase chain reaction (PCR). The results show that MgCl_2_ treatment for 8 h significantly reduces all epidermal differentiation mRNA expression levels (Figure 1A). Filaggrin, keratin 1, loricrin, and involucrin, used as epidermal differentiation markers, were also examined and all markers show suppression of expression (Figure 1B). In addition, significant suppression of keratin 1 and loricrin expressions by the MgCl_2_ treatment was also observed in normal human epidermal keratinocytes (NHEK). Filaggrin and involucrin showed a similar trend toward expression suppression, although the effects were not statistically significant. These results indicated that Mg^2+^ may inhibit epidermal differentiation.

### 2.2. Effects of Mg^2+^ on Cell Migration

A scratch assay was used to analyze the effect of both MgCl_2_ and MgSO_4_ on cell migration. The initial wound area was comparable, whereas the wound healing area increased in an MgCl_2_ concentration-dependent manner after 12 h (Figure 2A,B). In addition, to determine whether the migration-promoting ability was due to Mg^2+^, cell migration ability after MgSO_4_ treatment was also examined similarly. Treatment with MgSO_4_ significantly enhances cell migration, like MgCl_2_ (Figure 2C,D). The results indicate that Mg^2+^ enhances cell migration capacity.

### 2.3. Effect of Mg^2+^ on Cytoskeletal Rearrangement

As MgCl_2_ treatment enhanced the migratory capacity of HaCaT cells, the mechanism of this enhancement was investigated. Then, the effect of MgCl_2_ treatment on extracellular signal-regulated kinase (ERK) 1/2 phosphorylation was investigated using Western blotting. The results show that treatment with 5.8 mM MgCl_2_ markedly increases the phosphorylation levels of ERK1/2 (Figure 3A). Then, we analyzed MgCl_2_ treatment on cytoskeletal changes that occur in the early stages of migration by F-actin staining using phalloidin. This probe stains only polymerized actin (Figure 3B). The results show that MgCl_2_ treatment does not significantly change the intensity or morphology of the F-actin signal. Then, we performed a Western blotting analysis using an intracellular Ca^2+^ chelator to examine whether MgCl_2_ promotes the phosphorylation of ERK1/2 by altering intracellular Ca^2+^ influx. Under treatment with BAPTA-AM, MgCl_2_ significantly increased the phosphorylation of ERK (Figure 3C), suggesting that changes in Ca^2+^ influx may not exert MgCl_2_-induced activation of ERK. On the other hand, the downstream factor of MEK/ERK-induced cell proliferation *cyclin D1 (CCND1)* mRNA did not change by treatment with MgCl_2_ (Figure 3D), suggesting that MgCl_2_ may not promote cell proliferation. The above results indicate that MgCl_2_ treatment enhances ERK1/2 phosphorylation, an intracellular signaling factor required for cell migration, but may promote cell migration by a mechanism other than cytoskeletal reorganization by polymerized actin.

### 2.4. Effect of Mg^2+^ on MMPs Expression

Next, a real-time PCR analysis was performed to investigate the effect of MgCl_2_ treatment on the expression levels of MMP mRNA, and downstream factors of the ERK1/2 pathway. The results showed that MgCl_2_ treatment increased *MMP1* and *MMP7* mRNA expression levels (Figure 4A); the increase in *MMP7* mRNA expression levels by MgCl_2_ treatment was also observed in NHEK cells (Figure 4B). In contrast, *MMP1* mRNA expression levels were unchanged in NHEK cells; Western blotting analysis showed that MMP7 protein levels were increased by MgCl_2_ treatment (Figure 4C). Analysis of MMP7 protein localization by fluorescence immunostaining showed a stronger MMP7 protein signal with MgCl_2_ treatment (Figure 4D). Next, the MEK inhibitor was used to clarify the possibility of the MEK/ERK signal, a key pathway in cell signaling, regulating the MMP7 expression. Pretreatment with U0126, an MEK inhibitor, repressed the Mg^2+^-induced upregulation of MMP7 (Figure 4E). This suggests that the MEK/ERK signaling pathway is involved in the up regulation of MMP7, and its inhibition can prevent this process.

### 2.5. Effect of MMP7 Knockdown on Mg^2+^-Induced Cell Migration Promotion

To clarify whether MMP7 expression contributes to MgCl_2_-induced cell migration, MMP7 knockdown analysis was performed. In Western blotting analysis, MMP7 knockdown markedly reduced MMP7 protein expression (Figure 5A). Next, scratch assays in MMP7 knockdown cells were performed to investigate the effect of MgCl_2_ treatment on cell migration capacity. As a result, the cell migration-promoting effect of MgCl_2_ treatment seen in Figure 2 is not observed in MMP7 knockdown cells (Figure 5B). These results indicate that an increase in MMP7 protein expression mediates the cell migration-promoting effect of Mg^2+^.

## 3. Discussion

During the reconstructive phase of skin wound healing, epidermal keratinocytes are restored in functionally normal skin by appropriate differentiation [4,7,10]. Epidermal differentiation is regulated by various differentiation-related factors, including epidermal differentiation-promoting factors such as TP63, promoting epidermal stratification and cell adhesion [23,24]; DSG3, involved in growth inhibition and intercellular adhesion [25]; LCE1E [26] and FOSL2 [27], regulators of final epidermal differentiation and functioning in activating protein-1; and WNT5A, involved in inhibiting the WNT pathway, which is required for follicle formation and promotion of epidermal growth and differentiation. WNT5A, SEMA3C, and 5A are involved in nerve axon guidance and angiogenesis [28,29]. Filaggrin is a moisturizing factor of cornified envelopes [30]. Expression of the epidermal marker Keratin-1 is restricted to an intermediate stage of terminal differentiation [31]. Loricrin and involucrin are epidermal terminal differentiation factors [32,33]. The expression levels of all keratinocyte differentiation-related factors examined in the current study are decreased by MgCl_2_ treatment, suggesting that Mg^2+^ has an inhibitory contribution to epidermal re-differentiation in the late stages of skin wound healing.

During the proliferative phase of wound healing, epidermal keratinocyte migration, which occludes the wound edge, is an essential phenomenon [10]. In the present study, MgCl_2_ treatment enhances cell migration capacity in the scratch assay (Figure 2), suggesting that Mg^2+^ promotes wound healing during the early stages of healing when cell migration is required to seal the wound. Activation of ERK is essential for the migratory abilities of epidermal cells [34]. In the present study, phosphorylation of ERK is increased by MgCl_2_ treatment (Figure 3A). During cell migration, cytoskeletal reorganization occurs, forming leaf-like pseudopodia, filamentous pseudopodia, and stress fibers composed mainly of actin filaments [35]. ERK1/2 phosphorylation is known to be involved in this cytoskeletal reorganization [36]. However, F-actin is not changed by MgCl_2_ treatment (Figure 3B). The above results indicate that MgCl_2_ treatment enhances ERK1/2 phosphorylation, an intracellular signaling factor required for cell migration, but may promote cell migration by a mechanism other than cytoskeletal reorganization by polymerized actin. MMPs, proteases which are involved in extracellular matrix degradation, are known to play an important role in epidermal keratinocyte migration [9]. Multiple MMP subfamilies have also been reported to be downstream factors of the ERK1/2 pathway [37,38,39]. In the present study, MgCl_2_ treatment increases both protein and mRNA levels of MMP7 by activating the MEK/ERK pathway (Figure 4). Moreover, silencing MMP7 expression suppresses MgCl_2_ treatment-induced cell migration (Figure 5). These findings suggest that Mg^2+^ may promote keratinocyte migration by activating the MEK/ERK/MMP7 axis to promote wound healing. However, considering the effects of Mg^2+^ on differentiation, applying MgCl_2_ treatment to promote wound healing should not be limited to the entire wound healing process but only to the inflammatory phase when the inflammatory response is active and the proliferative phase when epidermal keratinocytes occlude the wound.

The physiological function of Mg^2+^ in cell differentiation differs between cell types. For example, Mg^2+^ promotes differentiation in osteoclasts and, conversely, inhibits differentiation in osteoblasts and skeletal muscle cells [40,41]. On the other hand, calcium ions (Ca^2+^) are differentiation-promoting factors in various tissues, including the epidermis [42]. In the epidermis, Mg^2+^ inhibits Ca^2+^ influx, so intracellular Ca^2+^ levels decrease with the increasing intracellular Mg^2+^ levels [43]. This report suggests that Mg^2+^ inhibits differentiation. It has also been reported that increased ERK1/2 phosphorylation in the epidermis suppresses epidermal differentiation [44], which is consistent with the suppressive effect of MgCl_2_ on the expression of epidermal differentiation-promoting factors observed in this study (Figure 1). The detailed mechanisms of ERK1/2 activation and epidermal differentiation suppression by Mg^2+^ are unknown and therefore require further analysis. In addition, as chronic enhancement of MMP expression has been reported to inhibit normal epidermal reconstitution and delay wound healing [45], the MMP7-enhancing effect of MgCl_2_ treatment may have an inhibitory effect on epidermal reconstruction.

MMP7 is a low-molecular-weight enzyme found in the uterus [46], and its physiological function in normal skin has not been previously reported. As MgCl_2_ promotes cell migration via increasing MMP7 expression (Figure 2 and Figure 5), MMP7 in the epidermis may be a dominant factor for physiologic keratinocyte migration. However, the mechanism of the detailed cell migration-promoting effect of MMP7 is still unclear. In a previous report using lung cells, MMP7 degraded a type of extracellular matrix syndecan-1, which is a substrate for MMP7, and reduced integrin affinity, thereby promoting cell migration [47]. MMP7 in the skin may also promote cell migration by a similar mechanism, and integrin expression and activity should also be investigated in the future. It has also been reported that Mg^2+^ enhances ERK1/2 phosphorylation [48]. We demonstrated that ERK1/2 phosphorylation enhances MMP7 expression (Figure 4E). However, it remains unknown how Mg^2+^ activates the MEK/ERK pathway and as such, further studies are needed.

Current treatments for skin wounds include debridement, electrical stimulation, and hyperbaric oxygen therapy, but there are issues such as invasiveness and the risk of oxygen intoxication [49]. Although oxygen was not involved in this study, MgCl_2_ treatment can be applied and is expected to have a minimally invasive therapeutic application. Trafermin, used to treat bedsores and ulcers, is essential for preparing human fibroblast growth factors. It cannot be applied to cancerous ulcers because of concerns that it causes the proliferation of cancer cells [50]. Further studies are needed to clarify its applicability to cancerous ulcers, as the growth-promoting effect of Mg^2+^ treatment has also been reported [16].

In patients with diabetes mellitus, advanced glycation end products (AGEs) accumulate due to hyperglycemia, blood flow reduces due to arteriosclerosis, and immune cells are dysfunctional, thereby resulting in delayed wound healing and ulceration of the feet [51,52]. Notably, AGE accumulation in patients with diabetes mellitus suppressed the expression levels of MMP7 in the epidermal stem cells [53], suggesting that MMP7 may be important for wound healing processes. In addition, hypomagnesemia is prevalent in patients with diabetes, and peripheral neuropathy and abnormal platelet activity due to hypomagnesemia are risk factors for diabetic foot lesions [54]. Based on the above reports, appropriate maintenance of blood Mg^2+^ levels in diabetic patients might be valuable in preventing and treating diabetic foot lesions [55].

## 4. Materials and Methods

### 4.1. Cell Cultures

Human epidermal keratinocyte derived HaCaT cells were allocated by Prof. Tadamichi Shimizu (Department of Dermatology, Faculty of Medicine, University of Toyama) and cultured in Dulbecco’s modified Eagle’s medium containing 5% fetal bovine serum (FBS), 0.14 mg/mL streptomycin sulfate, and 0.07 mg/mL penicillin G potassium at 37 °C with 5% CO_2_. Every 3–4 days, HaCaT cells were passaged using 0.02% ethylenediaminetetraacetic acid disodium salt (EDTA), phosphate-buffered saline (PBS), and 0.5% trypsin. NHEK cells were purchased from Kurabo (Tokyo, Japan). NHEK cells were cultured in a medium for keratinocytes (Lifeline Cell Technology, Frederick, MD, USA) at 37 °C with 5% CO_2_.

### 4.2. RNA Extraction and Reverse Transcription Reaction

RNAzol RT Reagent was used in accordance with the manufacturer’s instructions to extract total RNA from cells. Collected total RNA was dissolved in RNase-free water to make a total RNA sample. A reverse transcription reaction was then performed using a ReverTra Ace qPCR RT Kit (Toyobo, Osaka, Japan). The sample solution was incubated at 37 °C for 15 min, 50 °C for 15 min, and 95 °C for 5 min. After the reaction, the sample solution was used for real-time PCR.

### 4.3. Real-Time PCR

THUNDERBIRD Next SYBR qPCR Mix, primer pair, cDNA, and water were mixed. The PCR reaction was performed using Real-Time PCR Eco (As one, Tokyo, Japan) with three incubation steps as follows: (1) initial denaturation at 95 °C for 1 min; (2) annealing at 95 °C for 15 sec; and (3) elongation at 60 °C for 45 sec, all repeated for 40 cycles. After the PCR reaction, the growth and melting curves were analyzed to confirm non-specific reactions. When calibrated using some of the internal control genes, the changes in gene expression levels were similar. Therefore, β-actin was used as an internal standard for analysis. Primer sequences used in this study are presented in Table 1.

### 4.4. Scratch Assay

HaCaT cells were cultured in a 6-well plate to reach confluence. The cells were treated with 0.8–10.8 mM MgCl_2_ or 5.8 mM MgSO_4_. After 72 h incubation, the cells were incubated in serum-free medium for 16 h. After scratching with a pipette tip, bright field images of the cells were captured using a microscope (BZ810, Keyence, Osaka, Japan) at 0, 12, 18, and 24 h. A 0.5% FBS-containing medium was used during the recovery process to exclude the effects of cell proliferation. In the knockdown assay, HaCaT cells were transfected with a universal negative control and MMP7 siRNA using Lipofectamine RNAiMAX (Thermo Fisher Scientific, Waltham, MA, USA) for 48 h, and they were then used for the above scratch assay.

### 4.5. Sodium Dodecyl Sulfate (SDS)-Polyacrylamide Gel Electrophoresis (SDS-PAGE) and Western Blotting Analysis

HaCaT cells cultured in a 6-well plate were treated with 5.8 mM MgCl_2_. In the Ca^2+^ chelating experiment, 50 µM BAPTA-AM (Dojindo) was treated for 0.5 h before MgCl_2_ treatment. Cells were collected and added to a lysis buffer (150 mM NaCl, 1 mM EDTA, 20 mM Tris-HCl, 1% Triton X-100, 0.1% SDS, pH 7.4, protease inhibitor, phosphatase inhibitor cocktail, and 1% 2,2,2-trichloroethanol) and were disrupted by sonication (Bioruptor UCD-250, Cosmo Bio, Tokyo, Japan). After centrifugation at 6000× *g*, the supernatant was collected as a cell lysate. The cell lysate was prepared in sample buffer containing SDS and bromophenol blue, and the proteins were denatured at 99.9 °C for 5 min. Electrophoresis was performed using a 10% or 12.5% acrylamide gel at 25 mA for approximately 90 min. After electrophoresis, the proteins were irradiated with UVB at 1 J/cm^2^ in a UVP Crosslinker CL-1000M (Analytik Jena). The proteins were then transferred to polyvinylidene fluoride (PVDF) membranes in a semi-dry fashion at 100 mA for 60 min. After transfer, all protein bands were detected using LuminoGraph II (Atto, Tokyo, Japan) equipped with WUV-M20 UV Transilluminator. The membranes were immersed in a blocking solution (TBS-T containing 2% skim milk) and incubated at room temperature for 30 min. After removal of the blocking solution, the membrane was incubated with a primary antibody solution diluted (1:1000) in Solution 1 of Can Get Signal Immunoreaction Enhancer Solution (Toyobo) overnight at 4 °C. The following primary antibodies were used: anti-MMP7 antibody (A0695, ABclonal, MA, USA), anti-p-ERK1/2 antibody (sc-81492, Santa Cruz Biotechnology, TX, USA), and anti-ERK (p44/42 MAP kinase) antibody (#9102, Cell Signaling Technology, Danvers, MA, USA). The membranes were then incubated with horseradish peroxidase-conjugated secondary antibodies corresponding to the various primary antibodies for 90 min at room temperature. After treatment with the ECL detection reagents, the bands were detected using LuminoGraph II. Protein expression levels were corrected using total protein levels.

### 4.6. F-Actin Staining

The cells were stained with an F-actin-specific molecular probe, Phalloidin, to visualize the actin filaments in the early phase of cell migration, at 0, 1, and 6 h after scratching. After removal of the medium, cells were washed with PBS and fixed with 4% paraformaldehyde, permeabilized with 0.2% Triton X-100/PBS, and then incubated with the staining solution (100 nM rhodamine phalloidin and 572 nM DAPI) for 30 min. Fluorescence images of the stained cells were observed using Evos FL Auto 2 (Thermo Fisher Scientific).

### 4.7. Fluorescence Immunostaining

The cells were cultured in 6-well plates with a non-fluorescent cover glass placed at the bottom. After removal of the medium, the cells were washed with PBS, fixed with cold methanol at −30 °C, permeated with 0.2% Triton X-100/PBS at room temperature, blocked with 4% Block Ace for 30 min at room temperature, and reacted with the primary antibody (anti-MMP7 antibody) overnight at 4 °C. DAPI and the Alexa Fluor-546-labelled secondary antibody were reacted with the cells for 2 h, and the cover glass was then fixed on a glass slide with a mounting agent. Fluorescence immunostained cells were observed for fluorescence using a confocal laser microscope LSM 700 (Carl Zeiss, Jena, Germany). Images were captured using a 100× oil immersion objective on an inverted fluorescence microscope.

### 4.8. Statistical Analysis

Results are presented as mean ± standard error. Differences between groups were analyzed using one-way analysis of variance, and corrections for multiple comparisons were made using Dunnett’s test or Tukey’s multiple comparison test. Comparisons between the two groups were made using Student’s *t*-test. Statistical analyses were performed using the KaleidaGraph version 4.5.1 software (Synergy Software, Reading, PA, USA). Differences were considered to be significant at *p* < 0.05.

## 5. Conclusions

The mechanism of the wound healing-promoting action of MgCl_2_ and MgSO_4_ was investigated in the present study. The results revealed that MgCl_2_ inhibits epidermal differentiation and promotes cell migration through the MEK/ERK/MMP7 pathway. Our data suggest that Mg^2+^ is expected to be indicated for medical treatment of early-stage wounds.

## Figures and Tables

**Figure 1 ijms-25-04994-f001:**
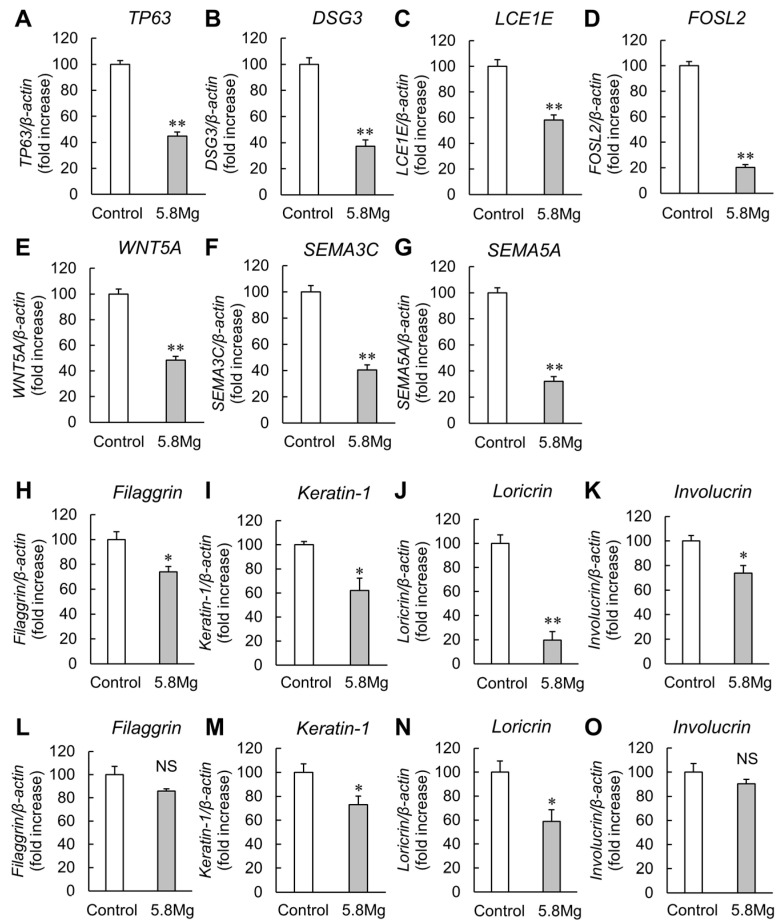
Effect of Mg^2+^ on the expression levels of keratinocyte differentiation-related genes. (**A**–**K**) HaCaT cells were cultured with DMEM containing 0.8 mM MgCl_2_ (Control) and 5.8 mM MgCl_2_ (5.8Mg) for 8 h. RT-PCR analysis showed the mRNA levels of (**A**) *TP63*, (**B**) *DSG3*, (**C**) *LCE1E*, (**D**) *FOSL2*, (**E**) *WNT5A*, (**F**) *SEMA3C*, (**G**) *SEMA5A*, (**H**) *filaggrin*, (**I**) *keratin-1*, (**J**) *loricrin*, (**K**) *involucrin*, and *β-actin*. *n* = 4. ** *p* < 0. 01, vs. Control. (**L**–**O**) NHEK cells were cultured with Control and 5.8 MgCl_2_ for 8 h. RT-PCR analysis showed the mRNA levels of (**L**) *filaggrin*, (**M**) *keratin-1*, (**N**) *loricrin*, (**O**) *involucrin*, and *β-actin*. *n* = 4. ** *p* < 0. 01, * *p* < 0.05, NS *p* > 0.05, vs. Control.

**Figure 2 ijms-25-04994-f002:**
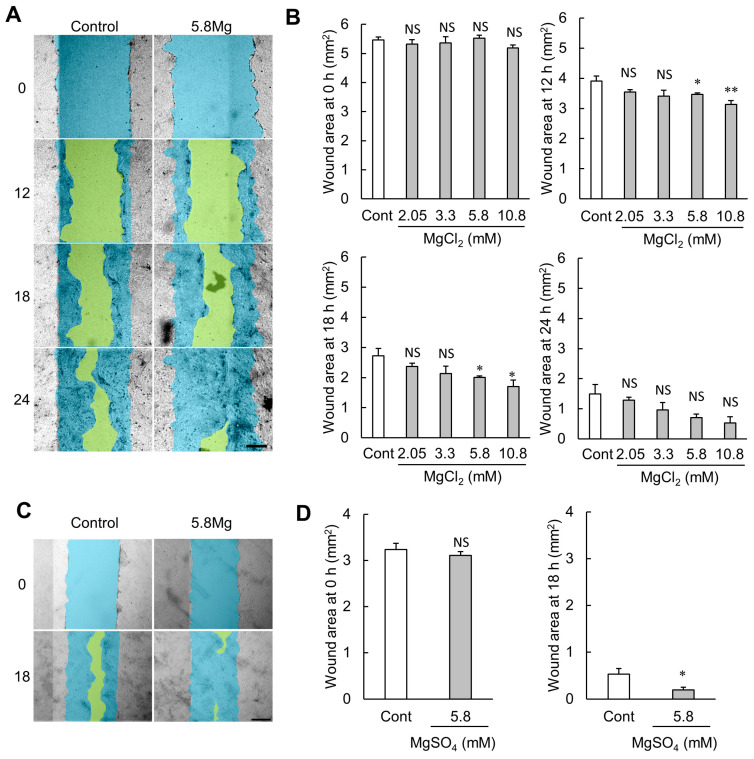
Effect of Mg^2+^ on cell migration. (**A**–**D**) HaCaT cells were cultured in 24 well plates until confluent and scratched with a pipette tip. Then, the cells were cultured in 0.5% FBS medium supplemented with 0.8–10.8 mM MgCl_2_ (**A**,**B**) or 5.8 mM MgSO_4_ (**C**,**D**). (**A**) The typical images of the cells with DMEM containing 0.8 mM MgCl_2_ and 5.8 mM MgCl_2_ are presented. (**B**) The scratched area was examined as a wound healing area (mm^2^) in each time point. *n* = 4. (**C**) Typical images of the cells with 5.8 mM MgSO_4_ are presented. (**D**) The scratched area was examined as a wound healing area (mm^2^) in 18 h. *n* = 4. ** *p* < 0.01, * *p* < 0.05, NS *p* > 0.05 vs. Control. Yellow shows the wounded area. Blue shows the initial wounded area.

**Figure 3 ijms-25-04994-f003:**
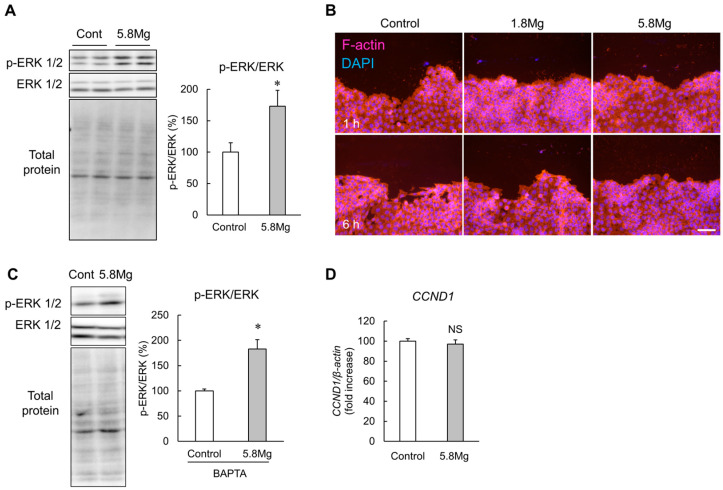
Effect of Mg^2+^ on cytoskeletal rearrangement. (**A**) HaCaT cells were cultured with DMEM containing 0.8 mM MgCl_2_ (Control) and 5.8 mM MgCl_2_ (5.8 Mg) for 1.5 h. Phosphorylated-ERK1/2 and total ERK1/2 protein levels were analyzed using Western blotting. *n* = 4–6. * *p* < 0.05 vs. Control. (**B**) HaCaT cells were cultured with Control and 5.8Mg for 0, 1, and 6 h after scratching with a pipette tip. The cells were observed using a fluorescence microscope after staining with Rhodamine Phalloidin (F-actin, red) and DAPI (nuclear, blue). Scale bar = 100 µm. (**C**) HaCaT cells were cultured with Control and 5.8Mg with 50 µM BAPTA-AM for 1.5 h. Phosphorylated-ERK1/2 and total ERK1/2 protein levels were analyzed using Western blotting. *n* = 4. * *p* < 0.05 vs. Control. (**D**) HaCaT cells were cultured with DMEM containing and 5.8 mM for 8 h. RT-PCR analysis showed the mRNA levels of *cyclin D1 (CCND1)* and *β-actin*. *n* = 4. NS *p* > 0.05 vs. Control.

**Figure 4 ijms-25-04994-f004:**
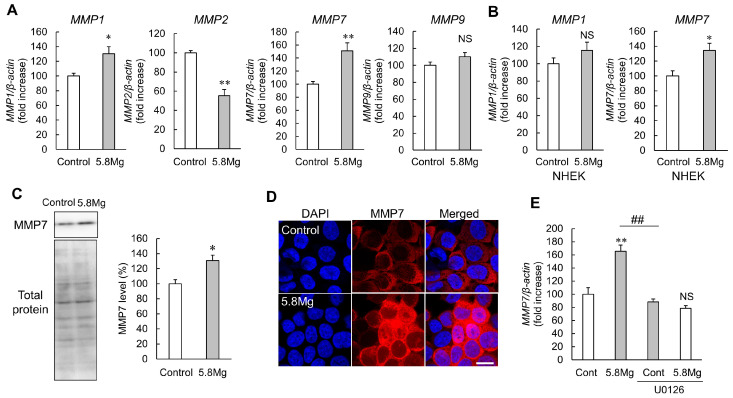
Effect of Mg^2+^ on MMPs expression. (**A**) HaCaT cells were cultured with DMEM containing 0.8 mM MgCl_2_ (Control) and 5.8 mM MgCl_2_ (5.8Mg) for 8 h. RT-PCR analysis showed the mRNA levels of *MMP1*, *MMP2*, *MMP7*, *MMP9*, and *β-actin*. (**B**) NHEK cells were cultured with Control and 5.8 Mg for 6 h. RT-PCR analysis showed the mRNA levels of *MMP1*, *MMP7*, and *β-actin*. (**C**) HaCaT cells were cultured in serum-free medium supplemented with Control and 5.8 Mg for 24 h. Western blotting analysis showed the protein level of MMP7. *n* = 4. ** *p* < 0.01, * *p* < 0.05, NS *p* > 0.05 vs. Control. (**D**) HaCaT cells were cultured on a cover glass with Control and 5.8Mg for 24 h. The cells were observed using confocal laser microscopy after staining with anti-MMP7 antibody (MMP7, red) and DAPI (nuclear, blue). Scale bar = 10 µm. (**E**) HaCaT cells were cultured with Control and 5.8 Mg with/or without 20 µM U0126 for 8 h. RT-PCR analysis showed the mRNA levels of *MMP7* and *β-actin*. *n* = 3–4. ** *p* < 0.01, NS *p* > 0.05 vs. Control. ## *p* < 0.01 vs. 5.8 Mg.

**Figure 5 ijms-25-04994-f005:**
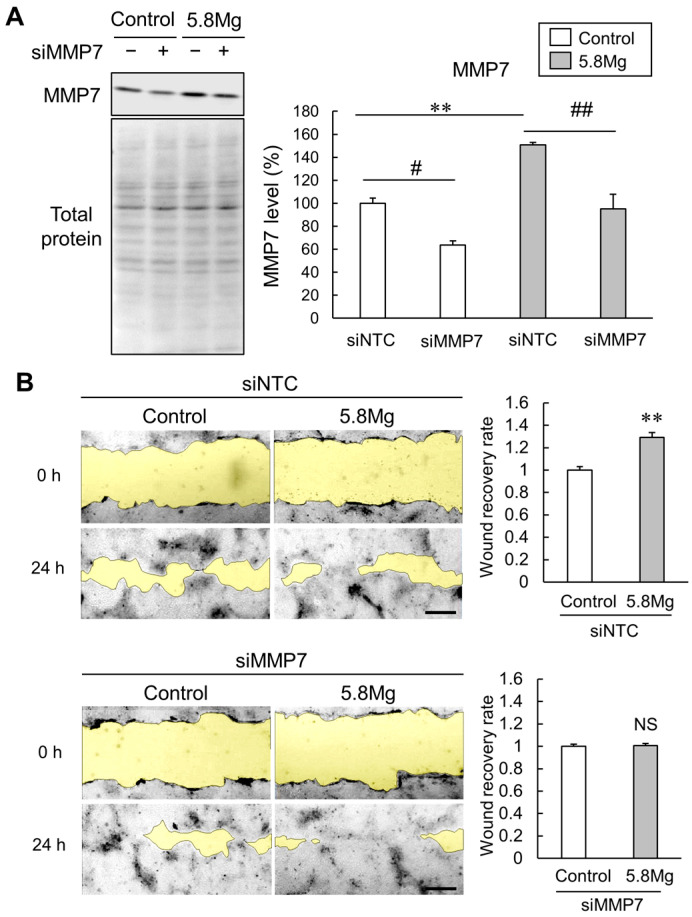
Effect of MMP7 knockdown on Mg^2+^-induced cell migration promotion. (**A**,**B**) HaCaT cells transfected with 50 nM siRNA for a negative control (siNTC) and with MMP7 (siMMP7) were cultured for 48 h to knockdown the expression of MMP7. (**A**) HaCaT cells were cultured with DMEM containing 0.8 mM MgCl_2_ (Control) and 5.8 mM MgCl_2_ (5.8 Mg) for 24 h after MMP7 knockdown. After collecting the cells, MMP7 protein levels were analyzed using Western blotting. *n* = 4. ** *p* < 0.01, vs. siNTC control. ## *p* < 0.01, # *p* < 0.05 vs. siNTC. (**B**) The cells were cultured in 0.5% FBS medium supplemented with Control and 5.8 Mg conditions and scratched with a pipette tip after MMP7 knockdown. The typical images of the cells are presented. Yellow shows the wounded area. The scratched area is examined for the wound recovery rate (ratio to control) at 24 h. *n* = 4. NS *p* > 0.05 vs. Control.

**Table 1 ijms-25-04994-t001:** Primer sequences used in real-time PCR.

Genes	Direction	Sequence (5′ → 3′)
*DSG3*	Forward	GACTCCTTCGGAAAGCAGCA
*DSG3*	Reverse	AAGCTTGGAGTTTCCTCCCG
*FOSL2*	Forward	GCCCAGTGTGCAAGATTAGC
*FOSL2*	Reverse	GGGCTCCTGTTTCACCACTA
*LCE1E*	Forward	TGAATAGCTGAGAGGTTCCAGC
*LCE1E*	Reverse	CAGCCATGGATCTGCAGAAG
*MMP1*	Forward	GGAGGAAATCTTGCTCAT
*MMP1*	Reverse	CTCAGAAAGAGCAGCATC
*MMP2*	Forward	ATGACGATGAGCTATGGACCTT
*MMP2*	Reverse	TCAGTGCAGCTGTTGTACTCCT
*MMP7*	Forward	TGTATGGGGAACTGCTGACA
*MMP7*	Reverse	GCGTTCATCCTCATCGAAGT
*MMP9*	Forward	TCTTCCAGTACCGAGAGAAAGC
*MMP9*	Reverse	GTCATAGGTCACGTAGCCCACT
*SEMA3C*	Forward	CAAAGATCCCACACACGGCT
*SEMA3C*	Reverse	ACTTGGTCCTCTGATCTCCTCC
*SEMA5A*	Forward	GTCTATACTTACTGCCAGCG
*SEMA5A*	Reverse	GTTAAATGCCTTGATGGCCTC
*TP63*	Forward	TTCTTAGCGAGGTTGGGCTG
*TP63*	Reverse	GATCGCATGTCGAAATTGCTC
*WNT5A*	Forward	CAATGAACCTACATAACAATGAAGC
*WNT5A*	Reverse	CAGCGGCAGTCTACTGACAT
*β-Actin*	Forward	CCTGAGGCACTCTTCCAGCCTT
*β-Actin*	Reverse	TGCGGATGTCCACGTCACACTTC
*Filaggrin*	Forward	GCAAGGTCAAGTCCAGGAGAA
*Filaggrin*	Reverse	CCCTCGGTTTCCACTGTCTC
*Keratin-1*	Forward	CAGACATGGGGATAGTGTGAGA
*Keratin-1*	Reverse	CAGGTCATTCAGCTTGTTCTTG
*Loricrin*	Forward	GGGCACCGATGGGCTTAG
*Loricrin*	Reverse	GGTAGGTTAAGACATGAAGGATTTGC
*Involucrin*	Forward	GGCCCTCAGATCGTCTCATA
*Involucrin*	Reverse	CACCCTCACCCCATTAAAGA
*CyclinD1*	Forward	TATTGCGCTGCTACCGTTGA
*CyclinD1*	Reverse	CCAATAGCAGCAAACAATGTGAAA

## Data Availability

Data are contained within the article.

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
