# Peer review of "Unraveling the Mechanisms Involved in the Beneficial Effects of Magnesium Treatment on Skin Wound Healing"

_ijms, 2024, doi:10.3390/ijms25094994_

Round 1
Reviewer 1 Report
Comments and Suggestions for Authors
In this report, the authors have investigated the effect magnesium ions wound healing in vitro and provided some mechanisms for this activity. Initially, authors show Mg ions (in the form of MgCl2) inhibits keratinocyte differentiation via assessing keratinocyte differentiation markers by qPCR. In scratchwound assays, addition of 5mM MgCl2, promoted wound closure compared to control. This wound accelerating effect of MgCl2 has been shown to be through the activation of ERK phosphorylation, which subsequently increases the synthesis of MMP7, an extracellular matrix degrading protease. The authors further showed that the pro-wound healing activity of MgCl2 could be inhibited, by treating cells with ERK inhibitor or MMP7 siRNA, further strengthening the ERK-MMP7 pathway, activated by MgCl2, involved in wound healing.
This manuscript is a follow up paper from the authors previous two publications wherein they have already shown the wound promoting effect of MgCl2 on keratinocytes through activation of hyaluronic acid synthesis and preventing UV-induced apoptosis. However, a novel mechanism is shown here (ERK-MMP7) which encourages this reviewer to recommend consideration of this manuscript. Some concerns are raised, which need to be addressed for publication.
1. The media used for keratinocyte culture (DMEM) has an intrinsic concentration of MgCl2 at 0.8 to 1mM. Do the authors consider this while adding 5mM MgCl2 or is it on top of this intrinsic concentration?
2. Though the effect of MgCl2 induced phosphoERK and MMP7 on its pro-wound healing activity, has been demonstrated using ERK inhibitors and MMP7 siRNA, experiment for the effect of MgCl2 using inhibitors is lacking. Chemical chelators such as EGTA/EDTA can be used to chelate MgCl2 and see if the wound healing activity is retained. This reviewer could understand the use of chelators (EGTA/EDTA) could have off-target effects (such as chelation of Ca2+ ions) which may confound the observations on scratchwound. However, these chelators could be used for the experiment showing MgCl2 could promote ERK phosphorylation within 1.5 h and see if EGTA/EDTA could inhibit MgCl2 induced phosphorylation of ERK. This is critical to see of MgCl2 exerts its effect via extracellular mechanisms or by entering into cells or by inhibiting Ca2+ influx.
3. A linear graphical model (depicting MgCl2 activating phosphoERK which induces MMP7 for wound healing) would be useful.
4. Would MgSO4 exert a similar effect like MgCl2 in scratchwound experiments?
5. This manuscript has to be thoroughly proof read by a native English speaker to enhance clarity.
6. The title can be changed to multivalent since ambivalent actually means having contradictory functions wherein MgCl2, inhibits differentiation and promotes wound healing via ERK-MMP7 axis, both of which are interlinked to each other in acute wound healing.
Comments on the Quality of English Language
English writing quality needs to be improved. Sentences can be improved and one sentence is not confirmative in its tone, such as
line no: 40, " cellular response called " wound healing" appears to maintain homeostasis
line no 33: Epidermis is the furthest layer containing keratinocytes (could be referred as "top most".
Author Response
Responses to Reviewer’s comments
Reviewer 1
In this report, the authors have investigated the effect magnesium ions wound healing in vitro and provided some mechanisms for this activity. Initially, authors show Mg ions (in the form of MgCl2) inhibits keratinocyte differentiation via assessing keratinocyte differentiation markers by qPCR. In scratchwound assays, addition of 5mM MgCl2, promoted wound closure compared to control. This wound accelerating effect of MgCl2 has been shown to be through the activation of ERK phosphorylation, which subsequently increases the synthesis of MMP7, an extracellular matrix degrading protease. The authors further showed that the pro-wound healing activity of MgCl2 could be inhibited, by treating cells with ERK inhibitor or MMP7 siRNA, further strengthening the ERK-MMP7 pathway, activated by MgCl2, involved in wound healing.
This manuscript is a follow up paper from the authors previous two publications wherein they have already shown the wound promoting effect of MgCl2 on keratinocytes through activation of hyaluronic acid synthesis and preventing UV-induced apoptosis. However, a novel mechanism is shown here (ERK-MMP7) which encourages this reviewer to recommend consideration of this manuscript. Some concerns are raised, which need to be addressed for publication.
Comment 1-1. The media used for keratinocyte culture (DMEM) has an intrinsic concentration of MgCl2 at 0.8 to 1mM. Do the authors consider this while adding 5mM MgCl2 or is it on top of this intrinsic concentration?
Response 1-1. We revised the control group and 5 mM MgCl2-treated as "DMEM containing 0.8 mM MgCl2 (Control)" and "5.8 mM MgCl2 (5.8Mg)", respectively.
Comment 1-2. Though the effect of MgCl2 induced phosphoERK and MMP7 on its pro-wound healing activity, has been demonstrated using ERK inhibitors and MMP7 siRNA, experiment for the effect of MgCl2 using inhibitors is lacking. Chemical chelators such as EGTA/EDTA can be used to chelate MgCl2 and see if the wound healing activity is retained. This reviewer could understand the use of chelators (EGTA/EDTA) could have off-target effects (such as chelation of Ca2+ ions) which may confound the observations on scratchwound. However, these chelators could be used for the experiment showing MgCl2 could promote ERK phosphorylation within 1.5 h and see if EGTA/EDTA could inhibit MgCl2 induced phosphorylation of ERK. This is critical to see of MgCl2 exerts its effect via extracellular mechanisms or by entering into cells or by inhibiting Ca2+ influx.
Response 1-2. According to reviewer suggestion, we performed western blotting analysis using MgCl2 with BAPTA-AM, an intracellular Ca2+ selective chelator. The data was added as Figure 3C. The following sentences were added in manuscript.
Then, we performed a western blotting analysis using an intracellular Ca2+ chelator to examine whether MgCl2 promotes ERK's phosphorylation by altering intracellular Ca2+ influx. Under treatment with BAPTA-AM, MgCl2 significantly increased the phosphorylation of ERK (Fig. 3C), suggesting that changes in Ca2+ influx may not exert MgCl2-induced activation of ERK. (p4, line 142)
Comment 1-3. A linear graphical model (depicting MgCl2 activating phosphoERK which induces MMP7 for wound healing) would be useful.
Response 1-3. According to reviewer suggestion, we revised graphical abstract to a liner graphical model.
Comment 1-4. Would MgSO4 exert a similar effect like MgCl2 in scratchwound experiments?
Response 1-4. According to reviewer suggestion, we performed a scratch assay with 5 mM MgSO4. The results were added as Figures 2C, D.
The following sentences were added in manuscript.
In addition, to determine whether the migration-promoting ability was due to Mg2+, cell migration ability after MgSO4 treatment was also examined similarly. Treatment with MgSO4 significantly enhanced cell migration, like MgCl2 (Fig. 2C, D). (p4, line 119)
Comment 1-5. This manuscript has to be thoroughly proof read by a native English speaker to enhance clarity.
Response 1-5. According to reviewer suggestion, our manuscript was checked by a professional English editing service (Medical English Service). Changed sentences were highlighted.
Comment 1-6. The title can be changed to multivalent since ambivalent actually means having contradictory functions wherein MgCl2, inhibits differentiation and promotes wound healing via ERK-MMP7 axis, both of which are interlinked to each other in acute wound healing.
Response 1-6. According to reviewer suggestion, title was changed to “Multivalent functions of magnesium ions on epidermal keratinocytes during skin wound healing”.
Comment 1-7.
Comments on the Quality of English Language
English writing quality needs to be improved. Sentences can be improved and one sentence is not confirmative in its tone, such as
line no: 40, " cellular response called " wound healing" appears to maintain homeostasis
line no 33: Epidermis is the furthest layer containing keratinocytes (could be referred as "top most".
Response 1-7. According to reviewer suggestion, our manuscript was checked by a professional English editing service (Medical English Service). Changed sentences were highlighted.
As reviewer suggested, the sentences were changed.
When the skin is damaged physically, a complex cellular response called "wound healing" is initiated to maintain homeostasis by restoring the injured area to its pre-injury state. (p1, line 31)
The epidermis is the thickest layer containing keratinocytes and melanocytes, and it comprises four layers including the basal layer, spinous layer, granular layer, and stratum corneum. (p1, line 38)
Reviewer 2 Report
Comments and Suggestions for Authors
Review of the manuscript entitled: Ambivalent functions of magnesium ions on epidermal
keratinocytes during skin wound healing. The manuscript is very interesting, but there are some elements that require improvement.
The introduction is properly prepared and describes the topic appropriately. In lines 54-56 maybe You should also mention elastin?
In the results section, it is completely unacceptable to provide references. In accordance with journal guidelines and good scientific practice, this must be corrected! The Authors mix the description of the results with a discussion. In the results section, we only describe our own results! To describe the background there is an introduction and to analyze the results there is a discussion sections. Results section are intended to describe the results in detail so that the reader can analyze them for themselves. The purpose of the discussion is for the authors to discuss and convince people of their theories.
As an example, lines between 97 to 115 should be moved to the discussion. And in result section Authors must results.
For example, “…8 h treatment of HaCaT cells by 5Mg decrease TP63 mRNA expression by 60.00% compared to control cells (Fig 1A). ” – correct entire result section and DESCRIBE results.
The discussion is fragmentary, it should be at least 2-3 times longer.
Antibody catalog numbers and dilutions should be added.
How did the authors choose b-actin as the reference gene? I know from experience that for studies related to collagen, elastin, metalloproteinases and matrix remodeling, b-actin is bad because it changes so much. Please describe what genes were considered and checked as controls. I understand that before the gene was selected, its stability was checked.
Author Response
Responses to Reviewer’s comments
Reviewer2
Comments and Suggestions for Authors
Review of the manuscript entitled: Ambivalent functions of magnesium ions on epidermal keratinocytes during skin wound healing. The manuscript is very interesting, but there are some elements that require improvement.
The introduction is properly prepared and describes the topic appropriately.
Comment 2-1. In lines 54-56 maybe You should also mention elastin?
Response 2-1. According to reviewer’s suggestion, we mentioned elastin in Introduction section.
Fibroblasts proliferate in the wound area, leading to the production of extracellular matrix components such as collagen, fibronectin, elastin, and hyaluronan and the formation of granulation tissue containing nerve fibers and capillaries. (p2, line 52)
Comment 2-2.
In the results section, it is completely unacceptable to provide references. In accordance with journal guidelines and good scientific practice, this must be corrected! The Authors mix the description of the results with a discussion. In the results section, we only describe our own results! To describe the background there is an introduction and to analyze the results there is a discussion sections. Results section are intended to describe the results in detail so that the reader can analyze them for themselves. The purpose of the discussion is for the authors to discuss and convince people of their theories.
As an example, lines between 97 to 115 should be moved to the discussion. And in result section Authors must results.
For example, “…8 h treatment of HaCaT cells by 5Mg decrease TP63 mRNA expression by 60.00% compared to control cells (Fig 1A). ” – correct entire result section and DESCRIBE results.
Response 2-2. Following the Reviewer's suggestion, we have rewritten the Results section so it does not contain citations. Changed sentences were highlighted.
Comment 2-3. The discussion is fragmentary, it should be at least 2-3 times longer.
Response 2-3. Following the Reviewer's suggestion, we have added sentences to the Discussion section and reorganized it.
Comment 2-4. Antibody catalog numbers and dilutions should be added.
Response 2-4. We added information about antibody catalog numbers and dilutions in Materials and Methods section.
After removal of the blocking solution, the membrane was reacted with in primary antibody solution diluted (1:1,000) in Solution 1 of Can Get Signal Immunoreaction Enhancer Solution (Toyobo) overnight at 4°C. (p11, line 360)
Primary antibodies used were following; anti-MMP7 antibody (A0695, ABclonal, MA, USA), anti-p-ERK1/2 antibody (sc-81492 ,Santa Cruz Biotechnology, TX, USA), anti-ERK (p44/42 MAP kinase) antibody (#9102, Cell Signaling Technology, MA, USA). (p11, line 363)
Comment 2-5. How did the authors choose b-actin as the reference gene? I know from experience that for studies related to collagen, elastin, metalloproteinases and matrix remodeling, b-actin is bad because it changes so much. Please describe what genes were considered and checked as controls. I understand that before the gene was selected, its stability was checked.
Response 2-5. Since β-actin and GAPDH were examined as the internal control genes and found to be comparable, β-actin was used in this study.
The following sentences were added in manuscript.
When calibrated using some of the internal control genes, the changes in gene expression levels were similar. Therefore, β-actin was used as an internal standard for analysis. (p9, line 328)
Round 2
Reviewer 2 Report
Comments and Suggestions for Authors
Thank you for considering my suggestions.
Author Response
Response to Academic Editor’s comments
Academic Editor Notes
The authors have appropriately addressed the reviewer's comments. However, there are some considerations that can improve the manuscript. Please, find my comments below:
Comment 1.
Regarding the title "Multivalent functions of magnesium ions on epidermal keratinocytes during skin wound healing" I find it very hard.
Since the authors exclusively investigated Mg effects on epidermal keratinocytes in vitro, the study cannot demonstrate Mg direct effects on skin wound healing. I would suggest changing the title to "Unraveling the mechanisms involved in the beneficial effects of magnesium treatment on skin wound healing" or others not directly related to skin wound healing, as various factors and cells can influence skin healing.
Response 1.
According to the suggestion, title was changed to “Unraveling the mechanisms involved in the beneficial effects of magnesium treatment on skin wound healing”.
Comment 2.
In the abstract, line 23, "...is desirable" should be replaced with "could be beneficial."
Response 2.
According to the suggestion, the sentence was changed to the following one.
Our results revealed that MgCl2 inhibits epidermal differentiation but promotes cell migration, suggesting that applying magnesium to the early wound healing process could be beneficial. (p1, Abstract)
Comment 3.
On page 3, line 90. The authors aim to investigate the effects of MgCl, but also study the effects of MgSO4. It might be more appropriate to indicate both or "magnesium treatment."
Response 3. According to the suggestion, the sentence was changed to the following one.
A scratch assay was used to analyze the effect of both MgCl2 and MgSO4 on cell migration. (p3, line )
Comment 4.
In Figure 1, I would recommend adding a single letter for every plot to refer properly in the text, since different cell lines and conditions are shown.
Response 4.
According to the suggestion, we added a single letter for every plot in Figure 1.
Comment 5.
On page 3, line 104. "significant expression suppression of keratin-1 and loricrin by MgCl2 treatment was also observed in normal human epidermal keratinocytes NHEK" contains grammatical errors.
Maybe it could be revised to "significant suppression of keratin-1 and loricrin expressions by MgCl2 treatment was also observed in normal human epidermal keratinocytes (NHEK)".
To avoid grammatical inconveniences, I would recommend english revision.
Response 5.
According to the suggestion, the sentence was revised to the following one.
In addition, significant suppression of keratin-1 and loricrin expressions by MgCl2 treatment was also observed in normal human epidermal keratinocytes (NHEK). (p3, line)
Comment 6.
In the legend of the figure 1. "After RNA extraction" should be changed to "RT-PCR analysis showed that...". This applies also to figure 4.
Response 6.
According to the suggestion, the sentence was changed to the sentences in Figures 1 and 4.
RT-PCR analysis showed the mRNA levels of MMP1, MMP2, MMP7, MMP9, and β-actin. (Figure1 legend)
Comment 7.
On page 4, line 137. "The results showed that treatment with 5 mM MgCl2 markedly increased". Did the author consider the Mg concentration in the culture medium? If not, they may refer to 5.8mM or 5mM supplementation respect to the Mg contained in the culture medium. This applies to the entire manuscript.
Response 7. According to the suggestion, this point was revised entire manuscript.
The results showed that treatment with 5.8 mM MgCl2 markedly increased the phosphorylation levels of ERK1/2 (Figure 3A). (p4, 2-3 line4)
HaCaT cells cultured in a 6-well plate were treated with 5.8 mM MgCl2. (p10, 4.5., line1)
Comment 8.
In the page 9, line 293. "advanced glycosylation end products" should be corrected to "advanced glycation end products"
Response 8.
According to the suggestion, the term "advanced glycosylation end products" changed to "advanced glycation end products". (p9, line4)
Comment 9.
On page 11, line 400. In the conclusion, the authors need to consider MgSO4, not just MgCl2, or refer to magnesium treatment. If magnesium supplementation inhibits differentiation and promotes cell migration, but could it also increase proliferation? In the methods, the authors mention that used 0.5% FBS to exclude proliferation in the scratch study; however, this may not be sufficient. As previous studies have shown that magnesium is involved in cell proliferation (cited by the authors on page 2, line 73) and activation of ERK pathway can also promote proliferation, the author may examine if this is like they suggest. To properly demonstrate that only cell migration is involved, they have to support their results with additional analysis of proliferation markers (e.g., PCNA, Cyclin D1) by western blot.
Response 9.
According to the suggestion, we conducted RT-PCR analysis to investigate the mRNA level of cyclin D1. The result of this analysis was added in Figure 3D. This data suggests that MgCl2 might not affect cell proliferation medium containing 0.5% FBS in the scratch assay. The following sentence is now included in the Results section.
On the other hand, the downstream factor of MEK/ERK-induced cell proliferation cyclin D1 (CCND1) mRNA did not change by treatment with MgCl2 (Figure 3D), suggesting that MgCl2 may not promote cell proliferation. (p4, 2.3., line14)
The mechanism of the wound healing-promoting action of MgCl2 and MgSO4 was investigated in the present study. (p11, 5. Conclusions, line1)